



# SeaConditions: a web and mobile service for safer professional and recreational activities in the Mediterranean Sea

G. Coppini[1], P. Marra[2], R. Lecci[1], N. Pinardi[1,3], S. Cretì[1], M. Scalas[2], L. Tedesco[2], A. D'Anca[4], L. Fazioli[5], A. Olita[5], G. Turrisi[1], C. Palazzo[1], G. Aloisio[1], S. Fiore[1], A. Bonaduce[1], Y. Kumkar[1], S. A. Ciliberti[1], I. Federico[1], G. Mannarini[1], P. Agostini[1], R. Bonarelli[1], S. Martinelli[1], G. Verri[1], L. Lusito[1], D. Rollo[2], A. Cavallo[2], A. Tumolo[2], T. Monacizzo[2], M. Spagnulo[2], R. Sorgente[5], A. Cucco[5], G. Quattrocchi[5], M. Tonani[6], M. Drudi[6], L. Panzera[1], A. Navarra[1], G. Negro[2]

[1] CMCC, Fondazione Centro Euro-Mediterraneo sui Cambiamenti Climatici, via Augusto Imperatore 16, 73100 Lecce, Italy
[2] Links S.p.a., Links Management and Technology, via R. Scotellaro, 73100 Lecce
[3] Università di Bologna, viale Berti-Pichat, 40126 Bologna, Italy
[4] CMCC, Fondazione Centro Euro–Mediterraneo sui Cambiamenti Climatici, Advanced Scientific Computing, Ecotekne Strada Provinciale per Arnesano, 73100 Lecce, Italy
[5] CNR-IAMC, Consiglio Nazionale delle Ricerche – Istituto per l'Ambiente Marino Costiero, Oristano, Italy
[6] INGV, Istituto Nazionale di Geofisica e Vulcanologia, Via Donato Creti 12, 40128 Bologna, Italy

*Correspondence to*: G. Coppini (giovanni.coppini@cmcc.it)

**Abstract.** The provision of reliable and timely information on the environmental conditions at sea to professional and recreational users is of strategic importance for their safety and for the optimal execution of their duties and activities. The capacity of the users of having the environmental information in due time and with the adequate accuracy in the marine and coastal environment can be defined as Sea Situational Awareness (SSA). Many activities are nowadays performed at sea also in absence of an adequate information related to the environmental meteorological and oceanographic conditions, in this situations the users will have a reduced capability in response, and this could lead to major emergencies including loss of lives and, large environmental disaster with consequent enormous damages on economy, society and ecosystems. In the framework of the TESSA project, new SSA services for the Mediterranean Sea were developed and in this paper we present one of them named SeaConditions a web and mobile application for the provision of meteorological and oceanographic observation and forecasting products.

Model forecasts and satellite products from operational services, such as ECMWF and CMEMS are visualized on SeaConditions. In addition, layers relative to bathymetry, sea level, ocean colour data (chl-a and water transparency) can be displayed. Since CMCC and CNR make available ocean forecasts at high spatial resolutions these are included in the version of SeaConditions presented.

SeaConditions aims to provide a user friendly experience with a fluid zoom capability, facilitating the proper display of data with different levels of details. SeaConditions consists of 'one stop shop' being a single point of access to interactive maps of quite different geophysical fields, delivering high quality information based on advanced oceanographic models.

The SeaConditions services are available through both web and mobile applications. The web application is available at the address www.sea-conditions.com

and is accessible to and compatible with present day browsers. Interoperability with GIS software is implemented. Users' feedback has been collected and considered for improving the service. The SeaConditions iOS and Android apps were downloaded so far (May 2016) by more than 105 000 users and more than 100.000 users have visited the web version.

## 1. Introduction

Sea Situational Awareness (SSA) in terms of an adequate dissemination of marine environmental data to the users and stakeholders is strategically important for management and safety purposes of Mediterranean Sea and its coastal areas. Situation in which low quality or absence of adequate information and consequent poor knowledge are available for operations at sea reduce the response capacity, leading to loss of lives and potential socio-economic damages. The SSA topic



is being addressed by TESSA, an industrial research project funded under the PON "Ricerca & Competitività 2007-2013" program of the Italian Ministry for Education, University and Research. TESSA is a joint effort of dissemination.

The first end-user oriented service developed in TESSA was SeaConditions (Lecci et al 2015): an open service providing ocean and weather forecasts, remote sensing data and bathymetry for the Mediterranean Sea, both for PC-based

and mobile channels. After the end of the project the Sea-Condition was exploited also as a commercial application in line the the industrial development aim of the funding framework which aims to increase the competitiveness of the Industry sector. The SeaConditions service is innovative because it aims to provide ocean forecasts to a wide community of users which includes many diverse type of users with different requirements. The high level of customization that can be reach in the MySeaCondition component of the service is a response of the very specific needs of some of the users such as repeated

bulletins in a specific location and only for selected variables (e.g. waves, wind, water temperature).

Other software or web portals similar to SeaConditions such as 'Weather4D' and 'Meteomed' do not offer the same integration with marine data such as currents, they do not high resolution model data, they do not offer the same zoom capability and they do not offer bathymetry as one of the product.

The service has been designed using a user-centred approach and through several technical meetings and large workshops

with tens of users we have analysed users' requirements, co-design the application, co-develop some of the features such as visualization of variables' palette and colours to be used, test the different versions and collected feedbacks. In the fields of weather services support applications SeaConditions is innovative because it aims to bring together 4 different types of information relevant in the SSA: 1) meteorological forecasts, 2) state of the art marine forecast 3) remote sensing observations, 4) bathymetry in one single one stop shop.

The paper is organized as follows. In Section 2 the operational chain is introduced subdivided in the two main components which are the operational centres and the SSA Platform. Section 3 presents the web portal, the mobile applications and the customizable version web application of MySeaConditions. Section 4 provides an overview of the interactions with testers and end users and Section 5 presents concluding remarks and a hint on future outlooks.

## 2. The operational chain

Figure 1 shows the SeaConditions operational chain comprising of five main steps:

1) the data acquisition of the generic forecast products. Thet stem from CMEMS (Copernicus Marine Environment Monitoring Service[1]) and from INGV (Istituto Nazionale di Geofisica e Vulcanologia) for the oceanographic forecasts, and from CNMCA (Centro Nazionale di Meteorologia e Climatologia Aeronautica) of the Italian Met-Office (Aeronautica Militare) for the meteorological forecasts;

2) the production of subregional model forecasts. They are available for the Adriatic Sea Forecasting System (AFS) produced by CMCC and the Tyrrhenian and Sicily Channel Forecasting System (TSCFS) produced by CNR-IAMC;

3) the post-processing by the CMCC of the products delivered by the CMEMS, AFS and TSCFS;

4) the map rendering process by Links S.p.A. performed on the dedicated SSA platform generates this information (e.g. maps and graphs) to the end-user. The data are presented as services in standard formats (like REST, TMS, WMTS) for the

integration into end-user applications. This activity is conducted

5) the delivery of the service to different end-user applications in a multi-channel mode (web and mobile, like tablet and smartphone). This activity is conducted by Links S.p.A..

---

[1] http://marine.copernicus.eu



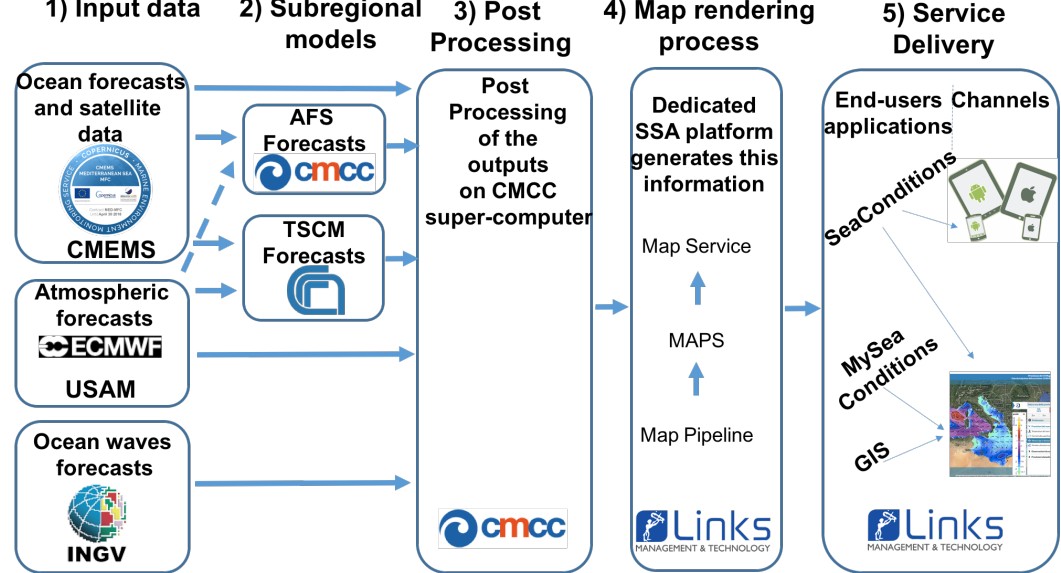

**Figure 1. The SeaConditions operational chain: the external data providers ECMWF, CMEMS and INGV feed the data production and collection centres, where high spatial resolution ocean forecast models are run. The output fields are then sent to the SSA platform for maps rendering, before they are made available to the end-user applications, through both the web and mobile channels.**

The following subsections describe of the components of the SeaConditions production and delivery chains mentioned above.

### 2.1 The Operational Centres (Input data, Sub-regional models and post-processing)

The SeaConditions operational chain starts everyday with the downloading of products delivered by CMEMS and CMCC and CNR-IAMC operational centres. CMCC perform the downloading and post-processing of the products to be then transferred to LINKS.

The IFS is the numerical model producing atmospheric data with a 1/8° x 1/8° horizontal resolution and is run and delivered by ECMWF. The Italian National Meteorological Centre (CNMCA) access the IFS data produced by ECMWF and makes them available to INGV. The IFS products used in SeaConditions consists of: 5-day forecast with a temporal resolution of 3h for the first three days and of 6h for the remaining 2 days. The variables used in SeaConditions are: temperature at 2m, total cloud cover, winds at 10m, mean sea level pressure and total precipitation. CMCC and CNR-IAMC download IFS data to

run their respective oceanographic models and CMCC uses them also to include them into SeaConditions as atmospheric products.

The oceanographic models products included in SeaConditions are the following:

- the CMESM MED-MFC products for ocean surface currents and temperature for the Mediterranean Sea in the framework of CMEMS;

- the MFS/INGV data for the waves for the Mediterranean Sea;

- the AFS data, for the ocean surface currents and temperature for the Adriatic Sea as produced by CMCC;

- the TSCFS data, for the ocean surface currents and temperature for the Tyrrhenian Sea and the Sicily Channel.



MFS is a coupled hydrodynamic-wave forecasting system developed and operational run by INGV since 2000 and now delivered in the framework of the CMEMS Med-MFC. The hydrodynamic model is based on Nucleus for European Modelling of the Ocean-Ocean Parallelise (NEMO-OPA) version 3.4 implemented in the Mediterranean Sea at 1/16° horizontal resolution and 72 unevenly spaced vertical levels (Oddo et al., 2009; Tonani et al., 2009). The wave dynamic is

based on WaveWatch-III (WW3) model (Tolman, 2009) implemented for the Mediterranean Sea. The model is forced at the surface by atmospheric data of analysis and forecasts produced by the ECMWF with a frequency of six hours and a spatial resolution of 0.25° (Tonani et al., 2008). The solutions are then corrected by the variational assimilation (based on a 3D-VAR scheme) of temperature and salinity vertical profiles and along track satellite sea level anomaly observations (Dobricic and Pinardi, 2008). MFS produces a 10-day forecast initialized by a simulation every day except Tuesday, when the analysis

is used instead of the simulation. The products are composed by 3D, hourly and daily mean fields of potential temperature, salinity, zonal and meridional velocity, and by 2D, hourly and daily mean fields of sea surface height and wave parameters, such as the significant wave height, the direction and the mean period.

AFS is a hydrodynamic forecasting system developed and maintained operational by INGV since April 2003. Starting from July 2013, it is maintained operational by the CMCC in Lecce (Italy). The hydrodynamic model is based on Princeton Ocean

Model (POM) implemented in the Adriatic Sea at 1/45° horizontal resolution and 31 sigma vertical levels (Oddo et al., 2005, 2006; Guarnieri et al. 2008). The domain encompasses the whole Adriatic basin and extends south of the Otranto channel into the northern Ionian Sea. The model is forced at the surface by atmospheric data of analysis and forecasts produced by ECMWF with a frequency of six hours and a spatial resolution of 0.25°. The river input into the basin has been implemented through river climatology (Raicich, 1994) for all the rivers except for the Po, which is a very important forcing for the

Adriatic Sea. The Po runoff implemented is on a daily basis. The initial and lateral boundary conditions for temperature, salinity and velocity come from MFS (Pinardi et al, 2003, Tonani et al, 2008). In order to have the best estimate of the state of the sea for the production of a new forecast, the daily cycle of AFS system is combined once a week with a weekly cycle taking place every Wednesday and providing the best initial conditions for the forecast. Everyday AFS produces 9-day forecast and the products are composed by 3D, hourly and daily mean fields of potential temperature, salinity, zonal and

meridional velocity, and by 2D, hourly and daily mean fields of sea surface height.

TSCFS is the evolution and extension of the Sicily Channel Regional Model (SCRM, Sorgente et al. 2003, 2011, Olita et al. 2007, 2012) that runs operational forecasts since 2003. It represents a new implementation of the Princeton Ocean Model at 1/48° of horizontal resolution and 31 vertical levels and provides a daily 5-days forecast of the 3D ocean hydrology (temperature, salinity) and circulation in the Central Mediterranean basin. Forecast fields are produced in "slave" mode,

meaning that they are re-initialized every day from the coarse model (MFS, Tonani et al., 2009), which also provides the boundaries for the system. At surfaces it receives 6-hourly forcing fields from ECMWF. Momentum and heat fluxes are computed interactively by the model through bulk formulae specific for the Mediterranean (Bignami et al., 1995; Castellari et al., 1998). The system also produces operational analyses through a 3D-variational Data Assimilation (DA) module since January 2015. The DA software assimilates Along Track sea level anomalies on a daily basis (Dobricic et al., 2008; Olita et

al., 2012).

The above mentioned products delivered by the oceanographic and atmospheric numerical models are everyday automatically downloaded from providers and stored into CMCC supercomputer (named *Okeanos*) as soon as they are available on the webserver of the providers. In the case of the high resolution ocean products AFS by CMCC, the data are directly available on *Okeanos*.

The first version of operational chain for SeaConditions is described in Lecci et al 2015 here we resume the main steps and highlight novelties:

1) The *scheduling phase*, which dispatches the data to the different phases implementing then a high parallelism for the computations;





2) The *pre-processing phase*, which deals with the time splitting and the conversion to CF netCDF format of the input data. At this stage, the operational chain subsamples the variables of interest for the requested temporal frames and for the surface level;

3) The *statistic phase*, where the statistics as minimum, maximum, mean and standard deviation are computed over the input data to then plot the proper palette in the rendering phase;

4) The *interpolation phase*, a demanding phase from the resources point of view that delivers at the end high spatial resolution data (150m). Atmospheric fields are provided over the original grid of 1/4° resolution (about 25km) except for 10m winds that are interpolated over a target grid of 2km resolution.

5) The *extrapolation phase*, ocean products are, in the coastal areas, extrapolated towards land using a procedure called SeaOverLand, which performs an extrapolation of the original data considering for each cell grid point an average of the 8 nearest values and then doing different iterations. This procedure optimally fills the gaps that remain between ocean model domain and the high-resolution coastline. Then a high-resolution mask is applied to remove the part of the extrapolated ocean data on land.

6) The *renaming and packing phase*, where, once transformed, all the data are packed using NetCDF libraries, with a decrease of their dimension to 1/10 of their original size.

The steps listed above are performed through a series of automated processing chains running 24h a day, 7 days a week (Lecci et al, 2015).

SeaConditions also delivers remote sensing products such as chlorophyll-a concentration and the transparency of seawater. These data are retrieved by satellite and are delivered on a grid of 1km spatial resolution by the CMEMS. Once available online, the data are automatically downloaded on Okeanos (CMCC) and transformed into the final output requested by SeaConditions.

In addition to remote sensing and model data SeaConditions provides information alos on sea level time series combined with 2 days sea level forecasts products for the Italian Seas. Sea level observations are delivered (36 stations in May 2015) by Rete Mareografica Nazionale (RMN) of Istituto Superiore per la Protezione e la Ricerca Ambientale (ISPRA) and CMCC combined them with an ensemble forecasting products of sea level.

In addition to the above-mentioned products, SeaConditions provides bathymetric information for the Mediterranean Sea provided by the Istituto Idrografico della Marina Militare (IIM). In particular bathymetry and awash rocks data are provided by the IIM and have been transformed by LINKS S.p.a. and CMCC in static layers that have been made available on SeaConditions[2].

## 2.2 The SSA Platform

The SSA Platform (Marra et al 2016; Lecci et al 2015) collects, transforms and provides forecast and observational products as information suitable for delivery across a variety of channels, like web and mobile.

The following are the core platform functional blocks: 1) the *Computing Grid* providing the platform with a highly scalable computing capability for processing forecast data and producing ready-to-use information (e.g. maps rendering); 2) the *Rendering Pipeline* arranging and supervising batch rendering jobs for processing NetCDF data; 3) the *Forecast Service* providing Application Programming Interfaces (APIs) for consuming the published forecast information (INSPIRE compliant), both as tiled maps and related metadata. Finally, it provides the on-demand rendering functionality which allows for on-the-fly single tile rendering, caching results on a local storage for later retrieval: this improves global performances while keeping the actual resources usage (e.g., disk space for storing maps) at the required minimum amount (Lecci et al, 2015).

---

[2] Starting from the end of TESSA project, information about sea level and bathymetry are no more supported.



## 3. The application

### 3.1 The portal

SeaConditions targets for the general public interested in knowing meteorological and oceanographic conditions at sea and provides a unique point of access to oceanographic and weather forecasts for the entire Mediterranean Sea. The strength of

SeaConditions is to integrate high quality meteorological and oceanographic forecast data with an innovative and easy-to-use approach for providing these data to a large number of consumers.

The first version of SeaConditions was published in July 2012 (Lecci et al, 2015), as a web portal (http://www.sea-conditions.com).

The version at the end of the TESSA project (May 2015) was providing the following data:

• Forecasting products:

a) ocean: surface temperature and currents, significant wave height and direction and wave period and direction;

b) atmosphere: air-temperature at 2 meter a.g.l., mean sea level pressure, total precipitation, total cloud coverage and winds at 10 meter a.g.l.

Once a day, a new set of forecasts for a time-span of four days and a half, are published;

• observation and forecast of sea level: observations are related to the past 8 days and forecasts to the following two days. Updated and published once a day;

• sea satellite observations: chlorophyll-a concentration, water transparency. Updated and published once a day;

• bathymetry and awash rocks. Static layers.

SeaConditions web portal interface is mainly composed of a map focused on the Mediterranean Sea and a menu for selecting data to display and forecast time steps and to set displaying options. The map takes up the main part of the screen: this allows the user an easier browsing of information about the forecast directly on the map.

Forecasts and observed data are displayed mainly as an information layer superimposed on a Google-map. Different kind of rendering has been studied and implemented in order to cope with the features of each kind of data: graphical rendering

includes maps colour shading and/or arrows and graphs representing the forecast trend.

Data expressed only by magnitude are rendered as shaded colour maps, while data expressed by magnitude and direction are displayed by both a shaded layer (for magnitude) and a vector layer in overlay (for direction). Figure 2 shows an example of map for the significant wave height and direction forecast data:

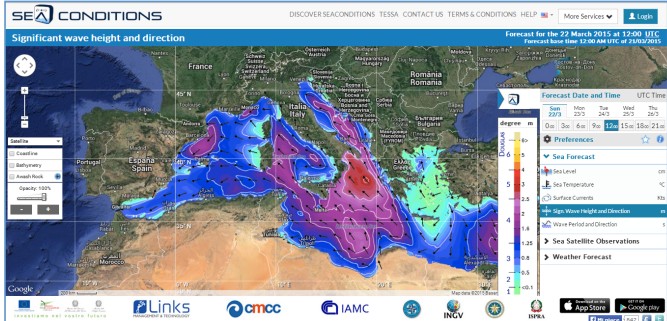


**Figure 2. SeaConditions web application displaying a significant wave height and direction map.**

Different kinds of arrows have been supported by the SSA platform on the base of the specific data to be represented: while wave directions are simple arrows, wind directions are displayed as arrows with barbs by embedding also information about wind magnitude. Concerning the currents, the streamlines have been used to better represent the circulation direction.

Every forecast data is associated to a reference scale displayed as a legend on the right side of the map. Depending on the kind of data considered, reference scales can be dynamic or static. Dynamic scales are defined every day by calculating the




minimum and maximum levels for the domain of the Mediterranean Sea: this applies to sea temperature, surface currents, wave period and air temperature. The other reference scales are fixed, based on state-of-the-art knowledge: this applies to wave height (Douglas scale), chlorophyll-a, water transparency, wind intensity (Beaufort scale), mean sea level pressure, precipitation and cloud cover.

5   The zooming capability allows the users to inspect specific areas with proper visualization through detailed maps. Figure 3 presents example of maximum zoom levels reached for waves high and directions (Figure 3.a), ocean surface temperature (Figure 3.b), wind (Figure 3.c) and surface ocean currents (Figure 3.d). Ocean surface currents products do not reach the same level of zoom reached with the other variables do to technical limitations in the plotting of streamlines curvilinear vector. CMCC and LINKS are working to overcome this problem.

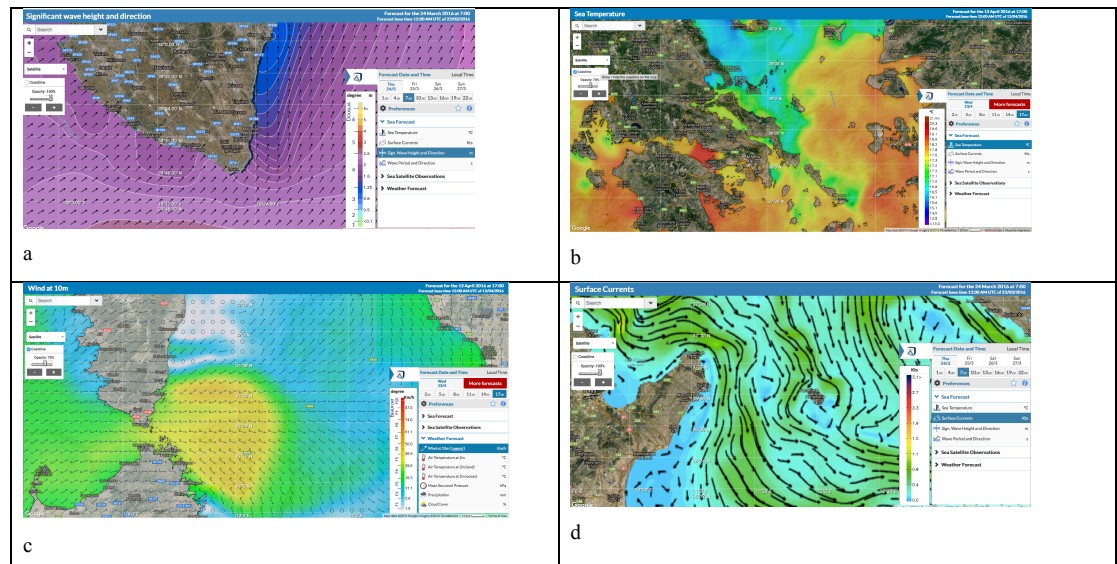

**Figure 3. Examples of SeaConditions products at various zoom levels: significant wave height and direction at level 11.. (Figure 3.a), surface sea temperature at level 7 (Figure 3.b), wind intensity and direction at level 10 (Figure 3.c), and surface sea currents at level 11 (Figure 3.d).**

15   Concerning sea level data, a different kind of rendering has been implemented: information are not displayed on a map, but as a set of points, one for each of the selected 22 tide gauge stations along the Italian coastline. For each station a threshold value is computed: when the sea level overcomes that specific value, the station is highlighted in red. By clicking every single station, it is possible to obtain sea level trend (both observations and forecasts) by means of a graph (figure 4).


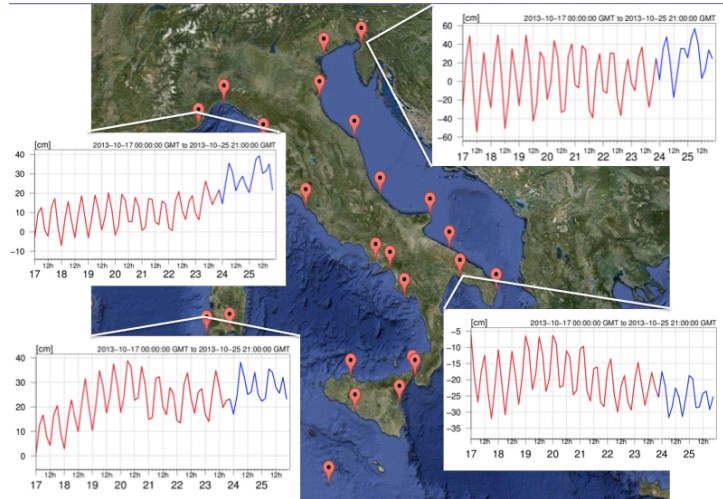

**Figure 4. Examples of SeaConditions sea level products in four different locations along the Italian coasts showing in red the time series of sea level observation and in blue the forecasts for the sea level next two days.**

In addition to maps (Figure 3), another way to explore forecast is by selecting one point on the map and by visualizing the trend for the forecast data in that specific point. Figure 5 shows an example of trend for surface currents forecast for a specific point of the map:

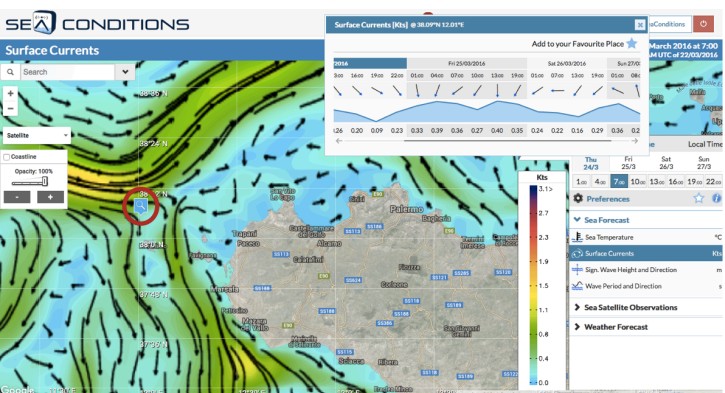

**Figure 5. SeaConditions web application– example of surface sea current map and (inset) timeseries of its intensity and direction variability in correspondence of the region highlighted with the red circle.**

High resolution model data (ocean surface temperature and currents) from AFS and TSCFS can be displayed on the web portal. The high resolution forecasts are displayed only at medium-high zoom level (after zoom 10 of google map) when the users has zoomed in and is starting to explore a sub-portion of the full domain. The borders of the 2 areas of the Adriatic Sea

and Northern Ionian and of the Tyrrhenian Sea and Sicily Channel are highlighted by a dashed white line on the screen so that the users know the domain of the high resolution data. Figure 6 illustrates an example of ocean surface currents in the central Mediterranean Sea with the sub-regional products displayed from the AFS and TSCFS data.



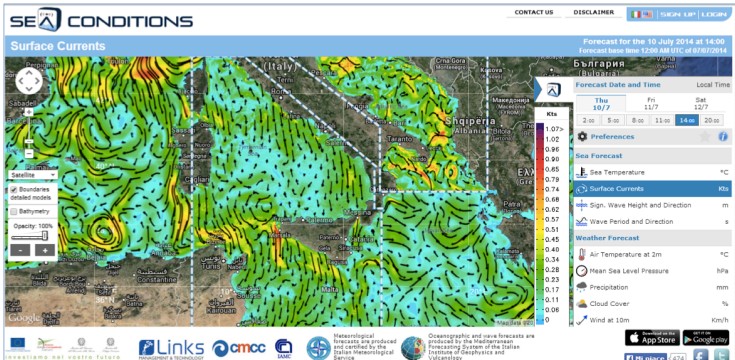

Figure 6. SeaConditions web application– example of ocean surface currents with the sub-regional products highlighted by the dashed white line.

Figure 7 presents the SeaConditions web application feature of showing bathymetry (Blue line). Data were provided by IIM as series of maps in GIS format, LINKS S.p.a. and CMCC have develop the methodology of merging together the different maps in unique layers to be displayed for the different google zoom levels.

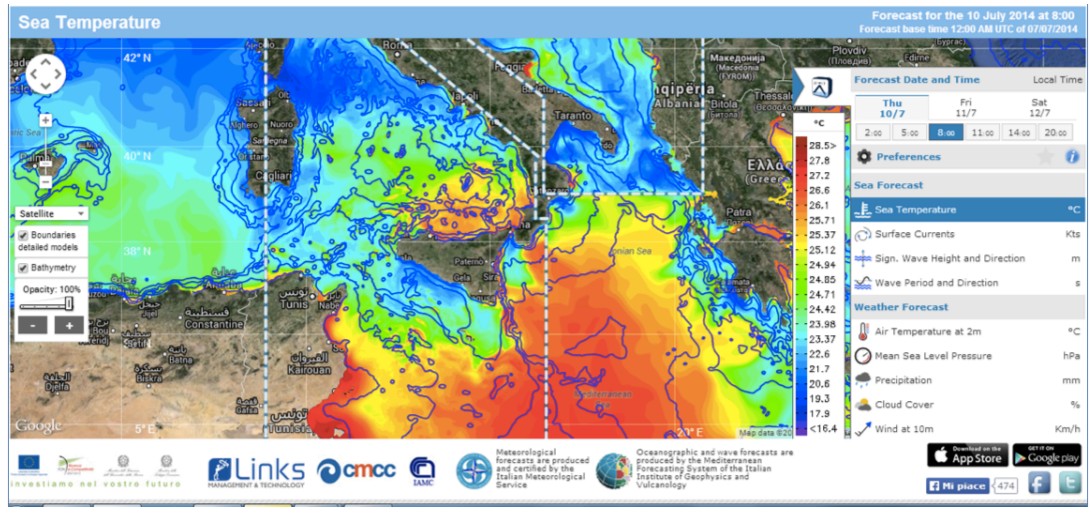

Figure 7. SeaConditions web application– example of bathymetry represented with blue lines.

15    Display options (i.e. the default forecast map, unit of measurement  can be set "once for all" by saving preferences (Lecci et al, 2015).

3.2 The application for mobile devices

SeaConditions is a multi-channel and multi-device service available both on the web and on mobile applications (Lecci et al,

20    2015) developed for smartphone and tablet, both for Android[3] and for IOS[4] devices.

_________________________

[3] https://play.google.com/store/apps/details?id=it.linksmt.tessa.scm
[4] https://itunes.apple.com/it/app/seaconditions/id737693930?mt=8




SeaConditions mobile has been built on a different paradigm but ensuring the same level of services compared to the web application. SeaConditions mobile has been designed by adopting a location based pattern: this way, in addition to the map views (as it is in SeaConditions web version), a navigation structure based on locations has been implemented. At this extent, specific views have been implemented for the provisioning of forecasts related to the user position ("around me") and

also to the users' favourite places (Lecci et al 2015).

In mobile version information are displayed not only as maps or graphs, but also in a more synthetic way by means of significant icons. In order to do this, a set of icons has been designed in order to represent every data related to the specific scale.

As regards the Android version, the "material design" guidelines[5] have been the main design principles followed and

adopted. Here are some examples of guidelines:

- sheets of paper in 3D space as the design metaphor;

- cards user interface and the preview/detail pattern;

- bold use of colour, typography, images and animations.

Figure 8 shows two example screens of the Android app:

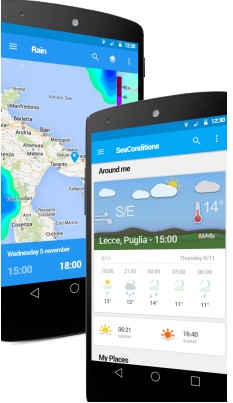

**Figure 8. SeaConditions Android mobile app – sample screens.**

The IOS version has been developed in the native language and is customized for iPhone and iPad.

Example of the screens from IOS applications are presented in Figure 9.

---

5 http://www.google.com/design/spec/material-design/introduction.html



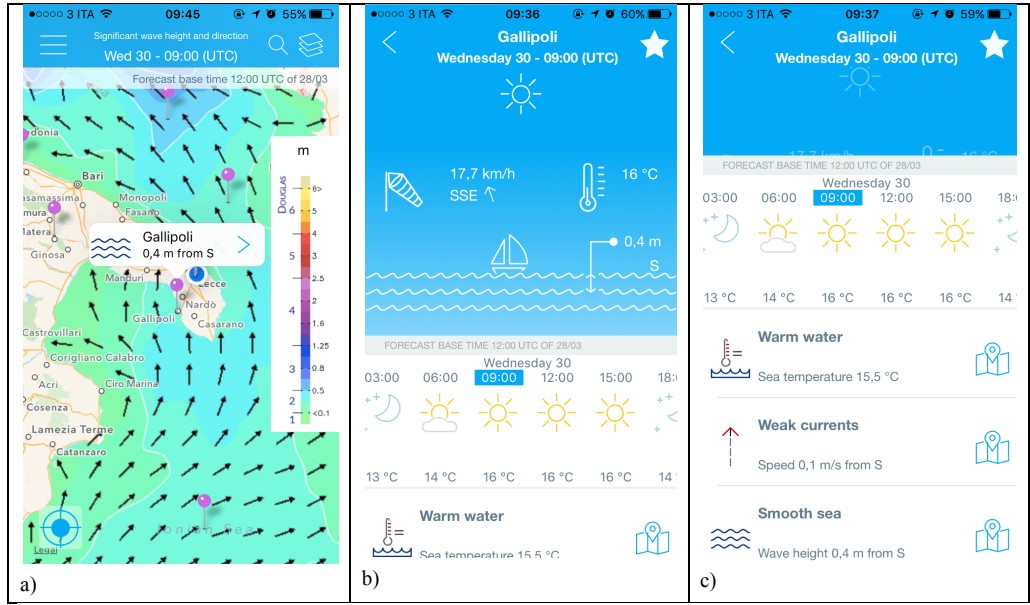

**Figure 9. SeaConditions iOS mobile app – sample screens: a) wave forecasts map with highlight of a sea location next to Gallipoli (40.05007 N, 17.96652 E) for which significant wave height and direction are reported; b) detailed bulletin for the selected location saved as a favourite place; c) yet another view of the bulletin with forecast information on more environmental fields.**

5   Continuos interaction with the users (e.g. workshops, supporting of sailing races, provision of questionnaires) provide the relevant information to keep SeaConditions under continuous improvement concerning both data and service quality.

### 3.3  MySeaConditions web application

MySeaConditions provides customized users-centred forecasts to users for any location of the Mediterranean Sea. After a registration to SeaConditions users can access MySeaConditions and benefit of additional services:

10   - registering favourite locations on the map;

- subscribing daily email forecast bulletins about favourite locations;

- subscribing alerts based on customized forecast thresholds for a specific location.

MySeaConditions (Figure 10) is only available on the web application of SeaCondition.



**Figure 10. MySeaConditions – example of the daily forecast bulletin generated for the location "Gallipoli" of Figure 8.**

## 4. Interactions with testers and end users

SeaConditions, as described in the previous paragraphs, has been developed adopting a user-centred approach, which allows to provide a service that matches the needs of the different users while maintaining easy-to-use and scientifically-sound features. Throughout the development of SeaConditions, many stakeholders, ranging from scientists to public authorities to different practitioners, have been involved and contacted for testing and feedback providing. The next subsection provides indications on the general framework that has been adopted for users interactions, while Subsection 4.2 reports some

interesting results of this long lasting and rewarding collaboration.

### 4.1 Framework for the interaction with users

The methodology adopted to involve users and provide continuous feedback for service improvement is composed of the following main steps:

- Identification of the users (e.g. public authorities, such as coastal guards, and private users, such as sailors, divers,

surfers and fishermen);
- Analysis of user requirements, obtained through living labs, participation to conferences/meetings/workshops, personal interactions during dedicated fairs and events;
- Experimentation planning, with definition of use cases and scenarios;
- User involvement that implied not only the experimentation of services in real scenarios, but also a survey

submitted to a wider list of users (companies involved in offshore activities, fishing consortia, marinas, shipping and generic transport companies);
- Collection, categorization and analysis of user feedbacks.

A first set of the service testing was performed with a restricted group of users, mainly from academia, and was focused on more functional tests about the usability of the User Interface (UI), the ability to facilitate the graphical output information

and the performance of the service in terms of zooming, map dragging and dropping, recovery storage output, etc.





Additional non-functional tests about UI usability and understanding, and performance concerning responsiveness and stability when subjected to particular workload were submitted to another group of key users, who were also substantially involved in the analysis of users requirements and correspondent features of SeaConditions. The list of these key users, defined Champion Users, included among others: the Italian Military Air Force, Regional (Apulia Region) Coast Guards,

Regional Agencies for Environmental Protection in Apulia and Emilia Romagna, the Port Authority of Bari, Regional (Apulia Region) Civil Protection, the sailing association Lega Navale and the 5-year sailing Mediterranea project crew.

A much wider set of users was involved in the evaluation of SeaConditions service through an online questionnaire and direct interviews, thanks to the collaboration with the Studio Valla SME. The questionnaire was composed of two main sections: the first one contained questions about the knowledge and habits of the users in consulting sites for forecasts, and

the availability of paying for this kind of services; the second section was focused instead on the investigation of interests of the users in the provided variables and their innovation potential, of the need of further information and a general evaluation of the usability of SeaConditions. The online questionnaire, developed within the platform SurveyMonkey, was mainly submitted to clubs of yachts and sport fishing, and surf and sub associations.

### 4.2 Some users results

A users database with a total number of 456 contacts was collected by Studio Valla, including the following categories: Offshore activities (62 contacts); Fishing (65 contacts); Commercial ports (26 contacts); Marinas (206 contacts); Shipping (ferries and boats) in Italy (55 contacts); generic transport in Europe, both tourist and commercial (Italy: 4 contacts; Turkey: 2 contacts, France: 5 contacts; Spain: 14 contacts, Greece: 11 contacts; Croatia: 3 contacts; Morocco: 1 contact; Malta: 2 contacts). Out of these users, around 50 questionnaires were received and Figure 11 reports some interesting results.

Generally, SeaConditions was well ranked by the respondents, with appreciation for the provided variables and their level of details. 70% of the respondents found the service quite good and the remaining 30% ranked it an optimal tool. Many respondents declared to use the system quite often (between once a day and more than once a week) and to be ready also to pay for the same service. Nonetheless, some suggestions for further improvement of the system were also noted down.


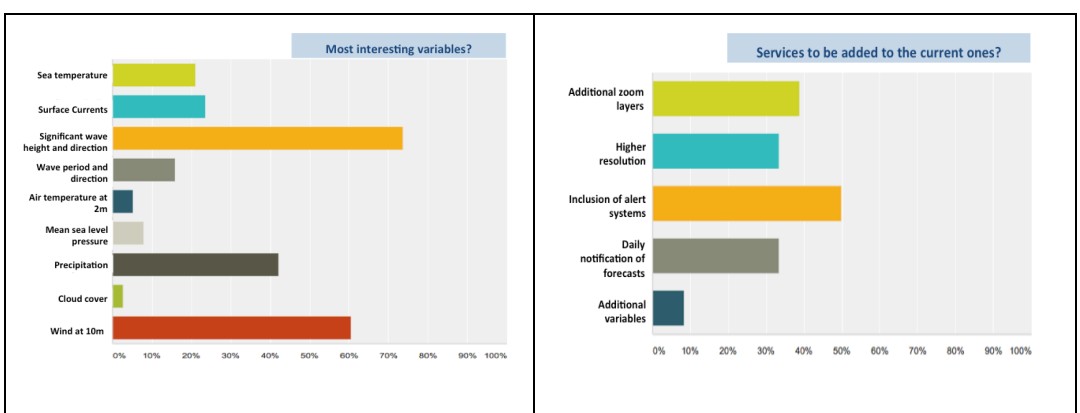



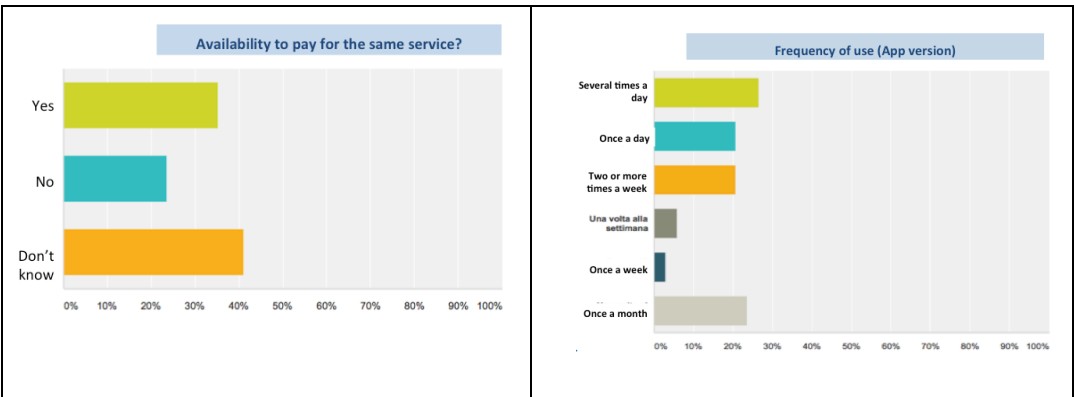

**Figure 11. Selected questions and answers by users to SeaConditions online questionnaire.**

An example of interaction with champion users is given by the Mediterranea sailing project
(www.progettomediterranea.com). This 5-year sailing throughout the Mediterranean Sea, the Black Sea and the Northern
Red Sea, Mediterranea wants to be a scientific floating laboratory, to study the sea, the climate change, the water and air
pollution, the marine biology, the currents and anything useful to preserve the ecological integrity of the sea. During the
navigation in the Mediterranean Sea, the crew used SeaConditions and reported through visuals and written articles their
experience with the service use. The videos and articles focused mainly on how the forecasts provided by SeaConditions
were matching the actual observations of the crew on the sea during the navigation. A feedback from Mediterranea is
summarized by the following statement of a member of the crew: "*A good system, with multiple choices in order to know the
real weather condition and forecast, easily usable and with a smart design with arrows and colors. A great mix of sources
that let the user trust the data and the projections. Quite good for sailors due to the accurate list of parameters to be
consulted. This is one of the few systems that gives period of the waves, so important in the route definition. SeaConditions,
in the North Aegean Sea, was a great help for studying the routes and navigate them.*"

## 5. Conclusions and outlook

SeaConditions represents today a new approach to the timely and reliable dissemination of weather-oceanographic
forecasting products to users for improving their sea situational awareness in the Mediterranean Sea.

SeaConditions is the result of a continuous users involvement: many different champion users have been selected to
participate in the testing phase providing relevant feedbacks for the service improvement. From the first publication of
SeaConditions, more than 100 000 people visited the website application and more than 105 000 users downloaded the two
mobile apps: users comments are in average very positive both as regards service usability and forecast accuracy. A set of
users was met during the project through dedicated meetings and workshops and interviewed through specific web
questionnaires. The Users of SeaConditions are professional actors using the service for their daily working activity (e.g.
mariners, fishermen, coast guards, navy), users practicing sport activities (e.g. sailing, diving, surf), including those that does
sports at professional level, and large public which visit the sea for recreational activities, including the bathing and touristic
activities. The interaction and dialogue with the so diversified population of users were challenging, they represented and
important investment during TESSA project, and they showed that new technology for SSA is needed in the touristic,
recreational and professional activities at sea. The important feedbacks provided by the users were first used to co-design
and co-develop SeaConditions and then at a later stage to verify and test all the different version of the Seaconditions portal
and apps. Based on user requirements, some future activities have been identified, such as the enlargement to other regions
out of Mediterranean Sea and the increase of number of products update within the same day.



SeaConditions represent from the scientific and technical point of view an important step forward in the visualization and exploitation of operational oceanography products for sea situational awareness. The experience of SeaConditions highlights that research and development activities should as much as possible aim to converge towards operational applications so that their results and finding are tested in operational mode and in real case scenarios by the users, often situations in which weak

components of the systems and bugs are highlighted. The operational testing also helps to demonstrate the effectiveness and importance of research results for supporting societal challenges and in our framework the benefit for maritime safety and for protection and management of marine environment.

**Acknowledgments**

This work was performed in the framework of the TESSA Project PON01_02823 supported by PON Ricerca & Competitività 2007-2013  cofunded by UE (Fondo Europeo di sviluppo regionale), MIUR (Ministero Italiano dell'Università e della Ricerca), and MSE (Ministero dello Sviluppo Economico). We thank Copernicus Marine Environment Monitoring

Service - CMEMS for the provision of the ocean forecasting products, Istituto Idrografico Marina Militare for the provision of the bathymetry data, Servizio Meteorologico Aeronautica Militare Italiano for the provision of the weather forecasting products and Istituto Superiore per la Protezione e la Ricerca Ambientale for the provision of the sea level data. We thank the Progetto Mediterranea for the fruitful scientific and technological collaboration and the support in dissemination of SeaConditions towards the Mediterraean users. We thank Studio Valla for the fruitful collaboration in the collection of

SeaConditions users' feedbacks.

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
