# Peer review of "SeaConditions: a web and mobile service for safer professional and recreational activities in the Mediterranean Sea"

_Natural Hazards and Earth System Sciences, 2016_

## Referee Comment (RC1) · Anonymous Referee #1 · 5 Aug 2016

General comments: The system presented by the authors combines in-situ data and forecasts and makes such information accessible to users to plan marine operations and activities that can be hazardous periodically. It is based on the best understanding of such hazards and can be considered state of the art.

Scientific Quality: In general, the scientific and technical approach is clear with some small exceptions. This reviewer's main concern is the extent to which the work presented herein overlaps with the previous publication by Lecci et al (2015). As this reference is not openly available it is difficult for this reviewer to assess the extent of an overlap in content (if any). The editor should satisfy themselves that the overlap is minimised. The reviewer welcomes the authors intent to publish the results to the wider

community.

Presentation Quality: Overall, the manuscript is well presented. Specific comments on language, figures etc are included later in this review.

Specific comments: This reviewer does not propose suggested changes to grammar, syntax etc but hopes that this will be picked up in the editing/proof reading process in more detail.

Page 2, line 6: "industry sector" should be replaced with "marine/maritime sector(s)". Line 11: not sure if "competitor" products should be mentioned. Editor should clarify journal policy in that respect. Line 35: "This activity….". Sentence is incomplete.

Page 3: Figure 1: USAM is not defined in the text or in the figure caption. Line 14: The nature/role of "LINKS" should be defined e.g. SME, research institute.

Page 4: Lines 2-35: A figure or table summarising the model extents and attributes would better serve the manuscript rather than the lengthy definition(s) provided. Line 20: A reference to or decsription of how river run-off is implemented would be helpful. Line 39: "Okeanos" is not defined

Page 5: Line 33: "channels" should be changed to "mechanisms" or "platforms".

Page 6: Line 14: " for days and a half" should be changed to "4.5 days". Line 32: "on the base of.." should be changed to "depending on the…."

Page 7: Line 6: "high" should be changed to "height".

Page 9: Figure 7: The intent is to show bathymetry. Suggest that temperature layer is removed to emphasise the bathymetry layer, hard to discern otherwise.

Page 10: Lines 9-12: The meaning of the first two bullet points are not clear and should be clarified. Figure 8: The images should be flattened and expanded to enhance readability.

Page 11: Lines 11 and 12: should be amended to "subscribing to" on both lines

Page 12: Line 6: " to provide a service that" should be changed to "the provision of…" Line 16: "living labs" should be explained or a reference provided Line 18: "experimentation" should be changed to "testing"

Page 13: Figure 11: The labels are difficult to read, this should be resolved before final publication.

Page 14: Many grammatical and spelling errors in "5. Conclusions and outlook"

---

## Referee Comment (RC2) · Anonymous Referee #2 · 23 Sep 2016

The ms provides the description of a new online, both via web and mobile, user friendly system to access and to visualize the Mediterranean ocean forecasting data provided by : CMEMS, sub-regional ocean systems, ECMWF and other complimentary sea data, in order to assist the end user and to strength the safety of mariners.

SeaConditions is an excellent tool to retrieve and visualize operationally the CMESM met-ocean data in the Mediterranean Sea.

However, it was desirable to provide in introduction a short description for 2-3 previous "similar" systems and their disadvantages compared to SeaConditions tool.

To my knowledge SeaConditions is well received by end users and not only and the

methodology behind it, can be adapted for other down streaming applications, such as oil spill and floating objects predictions.

---

## Author Comment (AC1) · 19 Dec 2016

Replies to Referee 1

Thank you very much for your very relevant comments. In the following text we present your comments followed by our answers and the modification that we propose following your comments to the final variant of the manuscript.

Referee's Comment 1 General comments: The system presented by the authors combines in-situ data and forecasts and makes such information accessible to users to plan marine operations and activities that can be hazardous periodically. It is based on the best understanding of such hazards and can be considered state of the art.

[Figure]

Authors' answer No comment.

Referee's Comment 2 Scientific Quality: In general, the scientific and technical approach is clear with some small exceptions. This reviewer's main concern is the extent to which the work presented herein overlaps with the previous publication by Lecci et al (2015). As this reference is not openly available it is difficult for this reviewer to assess the extent of an overlap in content (if any). The editor should satisfy themselves that the overlap is minimised. The reviewer welcomes the authors intent to publish the results to the wider C1 NHESSD Interactive comment Printer-friendly version Discussion paper community. Presentation Quality: Overall, the manuscript is well presented. Specific comments on language, figures etc are included later in this review.

Authors' answer The editor raised this issue in a previous version of the paper before submission. The paper has been extensively revised before final submission in order to minimize the overlaps with pervious publication Leccii et al (2015). Following the comment of the Referee we propose to add the following phrase in line 4 page 2: ". . .and it was presented in its preliminary version in Lecci et al 2015, this paper presents the final version of Sea-Conditions at the end of TESSA with a comprehensive overview of all components and of the different applications web and mobile."

Referee's Comment 3 Specific comments: This reviewer does not propose suggested changes to grammar, syntax etc but hopes that this will be picked up in the editing/proof reading process in more detail.

Authors' answer Several revisions of grammars are proposed in the text.

Referee's Comment 4 Page 2, line 6: "industry sector" should be replaced with "marine/maritime sector(s)". Line 11: not sure if "competitor" products should be mentioned. Editor should clarify journal policy in that respect. Line 35: "This activity. . ..". Sentence is incomplete.

Authors' answer Following the referee comment we have corrected as suggested in-

cluding the "marine/maritime sector(s). We propose to keep the phrase on different but similar products available on the web to highlith the differences and the original approach of SeaConditions. We have completed the sentence in line 35 adding "by Links S.p.A.."

Referee's Comment 5 Page 3: Figure 1: USAM is not defined in the text or in the figure caption. Line 14: The nature/role of "LINKS" should be defined e.g. SME, research institute.

Authors' answer We agreed with the Referee's comment and we have substituted the acronym USAM with IT Met Office and we have specify the nature of LINKS that is an SME.

Referee's Comment 6 Page 4: Lines 2-35: A figure or table summarizing the model extents and attributes would better serve the manuscript rather than the lengthy definition(s) provided. Line 20: A reference to or decsription of how river run-off is implemented would be helpful. Line 39: "Okeanos" is not defined

Authors' answer We agree with the referee's comment referring to the fact that a synthetic table is needed to summarize the forecasting system main characteristics. We proposed to keep the text since it is providing relevant details. We have add the Table 1 presenting the main characteristics of the forecasting systems used by SeaConditions.

As proposed by the referee we have added the references Oddo et al., 2005, 2006; Guarnieri et al. 2008 to provide references for the description of how the river run-off is implemented in AFS.

Following the Referee comment on "Okeanos" we have remove the name "Okeanos" and we have substitute with a generic reference to the CMCC supercomputing cluster.

Referee's Comment 8 Page 5: Line 33: "channels" should be changed to "mechanisms" or "platforms". Authors' Answer Following the referee advices we have substituted the word "channels" with "mechanisms".

Referee's Comment 9 Page 6: Line 14: " for days and a half" should be changed to "4.5 days". Line 32: "on the base of.." should be changed to "depending on the. . .." Authors' Answer We agree with the referee suggestions and we have modified the text accordingly.

Referee's Comment 10 Page 7: Line 6: "high" should be changed to "height". Authors' Answer We agree with the referee suggestion and we have modified the text accordingly.

Referee's Comment 11 Page 9: Figure 7: The intent is to show bathymetry. Suggest that temperature layer is removed to emphasise the bathymetry layer, hard to discern otherwise Authors' Answer We agree with the referee suggestion and we have modified the figure accordingly.

Referee's Comment 12 Page 10: Lines 9-12: The meaning of the first two bullet points are not clear and should be clarified. Figure 8: The images should be flattened and expanded to enhance readability. Authors' Answer We agree with the referee suggestion related to the first two bullet points of page 10 lines 9-12 and we have modified the text as following: As regards the Android version, the main principles adopted in designing SeaConditions mobile have been the "material design" guidelines[1]. Material design is "a comprehensive guide for visual, motion, and interaction design across platforms and devices"; here are examples of guidelines that have been applied: - organization of the user interfaces as the design metaphor of sheets of paper in 3D space. This imply that interfaces will have a shadow effect; - organization of information in portions of interface ("cards") accessible by a preview/detail pattern. It means that for each card, there will be two versions: the one with only a subset of information (i.e. only a small image and a caption) and the other with the complete set (i.e. the big image and complete text); - use of intense colours, bold characters, images and animations for being more attractive.

As regards of Figure 8: we have substituted the images with flattened and expanded

ones.

Referee's Comment 13 Page 11: Lines 11 and 12: should be amended to "subscribing to" on both lines Authors' Answer Following the Referees comment we have change the text as suggested.

Referee's Comment 14 Page 12: Line 6: " to provide a service that" should be changed to "the provision of. . ." Line 16: "living labs" should be explained or a reference provided Line 18: "experimentation" should be changed to "testing"

Authors' Answer Following the Referees comment we have change the text as suggested. Moreover we have added the following text to explain the Living lab concept: "AÂăliving labÂăis a research concept. A living lab is a user-centred,Âăopen-innovationÂăecosystem(Von Hippel, 1986; Chesbrough, 2003; Almirall et al 2011), integrating concurrent research and innovation processes within a public-private-people partnership (Pallot, 2009). The concept is based on a systematic user co-creation approach integrating research and innovation processes. These are integrated through the co-creation, exploration, experimentation and evaluation of innovative ideas, scenarios, concepts and related technological artefacts in real life use cases.ÂăIn our work we have applied the living lab methodology to the development of SeaConditions."

We have added the following references to the paper: Almirall, E., Wareham, J. (2011). Living Labs: Arbiters of Mid- and Ground- Level Innovation. Technology Analysis and Strategic Management, 23(1), 2011 pp. 87-102. http://dx.doi.org/10.1080/09537325.2011.537110 Chesbrough, H.W. (2003). Open Innovation: The new imperative for creating and profiting from technology. Boston: Harvard Business School Press. Boston (US). Von Hippel, E. (1986). Lead users: a source of novel product concepts. Management Science 32, 791–805. http://dx.doi.org/10.1287/mnsc.32.7.791

Referee's Comment 15 Page 13: Figure 11: The labels are difficult to read, this should be resolved before final publication

Authors' Answer Figures could be enlarged.

Referee's Comment 16 Page 14: Many grammatical and spelling errors in "5. Conclusions and outlook" Authors' Answer The grammatical and spelling errors have been corrected.

Marked up version of the manuscript is attached as supplement.

Please also note the supplement to this comment:
http://www.nat-hazards-earth-syst-sci-discuss.net/nhess-2016-176/nhess-2016-176-AC1-supplement.pdf

**Supplement:**

[revised manuscript text omitted]

operations at sea reduce the response capacity, leading to loss of lives and potential socio-economic damages. The SSA topic is being addressed by TESSA, an industrial research project funded under the PON "Ricerca & Competitività 2007-2013" program of the Italian Ministry for Education, University and Research. TESSA is a joint effort of dissemination.

The first end-user oriented service developed in TESSA was SeaConditions and it was presented in its preliminary version in

5 Lecci et al 2015, this paper presents the final version of Sea-Conditions at the end of TESSA with a comprehensive overview of all components and of the different applications web and mobile. Sea-Conditions is an open service providing ocean and weather forecasts, remote sensing data and bathymetry for the Mediterranean Sea, both for PC-based and mobile channels. After the end of the project the Sea-Condition was exploited also as a commercial application in line the the industrial development aim of the funding framework which aims to increase the competitiveness of the marine/maritime sector(s)

10 The SeaConditions service is innovative because it aims to provide ocean forecasts to a wide community of users which includes many diverse type of users with different requirements. The high level of customization that can be reach in the MySeaCondition component of the service is a response of the very specific needs of some of the users such as repeated bulletins in a specific location and only for selected variables (e.g. waves, wind, water temperature).

Other software or web portals similar to SeaConditions such as 'Weather4D' and 'Meteomed' do not offer the same

15 integration with marine data. Meteomed does not offer currents and high resolution model data. Both Weather4D and Meteomed do not offer the zoom capacity in the free version. Both of them do not offer bathymetry as one of the product. INGV has developed, since the begin of the Mediterranean Forecasting System (MFS), web interface to visualise the oceanographic products of the Mediterranean Sea (http://medforecast.bo.ingv.it/mfs-copernicus) developed in the different projects (MFSPP, MFSTEP, MYOCEANs and now CMEMS). The INGV website is more scientific oriented respect to

20 SeaConditions, it consists of static images without any zoom capability and does not provide the meteorological forecasts. Other many systems (e.g. Puertos forecasting system portus.puertos.es; CYCOFOS forecasting systems http://www.oceanography.ucy.ac.cy/cycofos/high-resolution.html), most of them part of MONGOOS (http://www.mongoos.eu/in-situ-and-forecasts), provide the access to visualization services of oceanographic forecasting products but all of them for sub-regional areas only and not for the entire Mediterranean sea, most of them with limited

25 zoom capability and without providing access to atmospheric data. Only few of the forecasting systems in the Mediterranean Sea provide access through mobile applications and only for sub-regional areas of the Mediterranean Sea (IMAR by Puertos available on Google store for the Western Mediterranean Sea).

The service has been designed using a user-centred approach and through several technical meetings and large workshops

30 with tens of users we have analysed users' requirements, co-design the application, co-develop some of the features such as visualization of variables' palette and colours to be used, test the different versions and collected feedbacks. In the fields of weather services support applications SeaConditions is innovative because it aims to bring together 4 different types of information relevant in the SSA: 1) meteorological forecasts, 2) state of the art marine forecast 3) remote sensing observations, 4) bathymetry in one single one stop shop.

35 The paper is organized as follows. In Section 2 the operational chain is introduced subdivided in the two main components which are the operational centres and the SSA Platform. Section 3 presents the web portal, the mobile applications and the customizable version web application of MySeaConditions. Section 4 provides an overview of the interactions with testers and end users and Section 5 presents concluding remarks and a hint on future outlooks.

**2. The operational chain**

40 Figure 1 shows the SeaConditions operational chain comprising of five main steps:

Field Code Changed

1) the data acquisition of the generic forecast products. Thet stem from CMEMS (Copernicus Marine Environment Monitoring Service[1]) and from INGV (Istituto Nazionale di Geofisica e Vulcanologia) for the oceanographic forecasts, and from CNMCA (Centro Nazionale di Meteorologia e Climatologia Aeronautica) of the Italian Met-Office (Aeronautica Militare) for the meteorological forecasts;

2) the production of subregional model forecasts. They are available for the Adriatic Sea Forecasting System (AFS) produced by CMCC and the Tyrrhenian and Sicily Channel Forecasting System (TSCFS) produced by CNR-IAMC;

3) the post-processing by the CMCC of the products delivered by the CMEMS, AFS and TSCFS;

4) the map rendering process by Links S.p.A. performed on the dedicated SSA platform generates this information (e.g. maps and graphs) to the end-user. The data are presented as services in standard formats (like REST, TMS, WMTS) for the integration into end-user applications. This activity is conducted by Links S.p.A..

5) the delivery of the service to different end-user applications in a multi-channel mode (web and mobile, like tablet and smartphone). This activity is conducted by Links S.p.A..

[Figure]

**Figure 1. The SeaConditions operational chain: the external data providers ECMWF, CMEMS and INGV feed the data production and collection centres, where high spatial resolution ocean forecast models are run. The output fields are then sent to the SSA platform for maps rendering, before they are made available to the end-user applications, through both the web and mobile channels.**

The following subsections describe of the components of the SeaConditions production and delivery chains mentioned above.
* * *
[1] http://marine.copernicus.eu

**2.1 The Operational Centres (Input data, Sub-regional models and post-processing)**

The SeaConditions operational chain starts everyday with the downloading of products delivered by CMEMS and CMCC and CNR-IAMC operational centres. CMCC perform the downloading and post-processing of the products to be then
5    transferred to LINKS S.p.A. (SME).

The IFS is the numerical model producing atmospheric data with a 1/8° x 1/8° horizontal resolution and is run and delivered by ECMWF. The Italian National Meteorological Centre (CNMCA) access the IFS data produced by ECMWF and makes them available to INGV. The IFS products used in SeaConditions consists of: 5-day forecast with a temporal resolution of 3h for the first three days and of 6h for the remaining 2 days. The variables used in SeaConditions are: temperature at 2m, total
10    cloud cover, winds at 10m, mean sea level pressure and total precipitation. CMCC and CNR-IAMC download IFS data to run their respective oceanographic models and CMCC uses them also to include them into SeaConditions as atmospheric products. The main characteristics of the forecasting systems used by SeaConditions are presented in table 1.

The oceanographic models products included in SeaConditions are the following:

-    the CMESM MED-MFC products for ocean surface currents and temperature for the Mediterranean Sea in the
15        framework of CMEMS;
-    the MFS/INGV data for the waves for the Mediterranean Sea;
-    the AFS data, for the ocean surface currents and temperature for the Adriatic Sea as produced by CMCC;
-    the TSCFS data, for the ocean surface currents and temperature for the Tyrrhenian Sea and the Sicily Channel.

| Forecasting system name | Model name | Main characteristics |
|---|---|---|
| CMESM MED-MFC (MFS) | NEMO-OPA version 3.4 | Horizontal resolution: 1/16° Vertical levels: 72 unevenly spaced |
| MFS/INGV waves | WaveWatch-III (WW3) | Horizontal resolution: 1/16° |
| IFS | IFS[2] | Horizontal resolution 1/4° |
| AFS | Princeton Ocean Model (POM) | Horizontal resolution: 1/45° Vertical levels: 31 sigma vertical levels |
| TSCFS | Princeton Ocean Model (POM) | Horizontal resolution: 1/48° Vertical levels: 31 sigma vertical levels |

20    Table 1: Main characteristics of the forecasting systems used by SeaConditions.

[revised manuscript text omitted]
 main principles adopted in designing SeaConditions mobile have been the "material design" guidelines[6]. Material design is "a comprehensive guide for visual, motion, and interaction design across platforms and devices"; here are examples of guidelines that have been applied:

- organization of the user interfaces as the design metaphor of sheets of paper in 3D space. This imply that interfaces will have a shadow effect;
* * *
4 https://play.google.com/store/apps/details?id=it.linksmt.tessa.scm
5 https://itunes.apple.com/it/app/seaconditions/id737693930?mt=8
6 http://www.google.com/design/spec/material-design/introduction.html

- organization of information in portions of interface ("cards") accessible by a preview/detail pattern. It means that for each card, there will be two versions: the one with only a subset of information (i.e. only a small image and a caption) and the other with the complete set (i.e. the big image and complete text);

- use of intense colours, bold characters, images and animations for being more attractive.

Figure 8 shows two example screens of the Android app:

[Figure]

[Figure]

**Figure 8. SeaConditions Android mobile app – sample screens.**

The IOS version has been developed in the native language and is customized for iPhone and iPad.
Example of the screens from IOS applications are presented in Figure 9.

[Figure]

[Figure]

**Figure 9. SeaConditions iOS mobile app – sample screens: a) wave forecasts map with highlight of a sea location next to Gallipoli (40.05007 N, 17.96652 E) for which significant wave height and direction are reported; b) detailed bulletin for the selected location saved as a favourite place; c) yet another view of the bulletin with forecast information on more environmental fields.**

Continuos interaction with the users (e.g. workshops, supporting of sailing races, provision of questionnaires) provide the relevant information to keep SeaConditions under continuous improvement concerning both data and service quality.

**3.3 MySeaConditions web application**

MySeaConditions provides customized users-centred forecasts to users for any location of the Mediterranean Sea. After a registration to SeaConditions users can access MySeaConditions and benefit of additional services:

- registering favourite locations on the map;

- subscribing to daily email forecast bulletins about favourite locations;

- subscribing to alerts based on customized forecast thresholds for a specific location.

MySeaConditions (Figure 10) is only available on the web application of SeaCondition.

**Figure 10. MySeaConditions – example of the daily forecast bulletin generated for the location "Gallipoli" of Figure 8.**

**4. Interactions with testers and end users**

SeaConditions, as described in the previous paragraphs, has been developed adopting a user-centred approach, which allows the provision of a service which matches the needs of the different users while maintaining easy-to-use and scientifically-sound features. Throughout the development of SeaConditions, many stakeholders, ranging from scientists to public authorities to different practitioners, have been involved and contacted for testing and feedback providing. The next subsection provides indications on the general framework that has been adopted for users interactions, while Subsection 4.2 reports some interesting results of this long lasting and rewarding collaboration.

**4.1 Framework for the interaction with users**

The methodology adopted to involve users and provide continuous feedback for service improvement is composed of the following main steps:

- Identification of the users (e.g. public authorities, such as coastal guards, and private users, such as sailors, divers, surfers and fishermen);

- Analysis of user requirements, obtained through living labs, participation to conferences/meetings/workshops, personal interactions during dedicated fairs and events;
- Experimentation planning, with definition of use cases and scenarios;
- User involvement that implied not only the experimentation of services in real scenarios, but also a survey submitted to a wider list of users (companies involved in offshore activities, fishing consortia, marinas, shipping and generic transport companies);
- Collection, categorization and analysis of user feedbacks.

A living lab is a research concept. A living lab is a user-centred, open-innovation ecosystem(Von Hippel, 1986; Chesbrough, 2003; Almirall et al 2011), integrating concurrent research and innovation processes within a public-private-people partnership.
The concept is based on a systematic user co-creation approach integrating research and innovation processes. These are integrated through the co-creation, exploration, experimentation and evaluation of innovative ideas, scenarios, concepts and related technological artefacts in real life use cases. In our work we have applied the living lab methodology to the development of SeaConditions.

[revised manuscript text omitted]

Almirall, E., Wareham, J. (2011). Living Labs: Arbiters of Mid- and Ground- Level Innovation. Technology Analysis and Strategic Management, 23(1), 2011 pp. 87-102. http://dx.doi.org/10.1080/09537325.2011.537110

Bignami F., Marullo S., Santoleri R., Schiano M.E. (1995), "Longwave radiation budget in the Mediterranean Sea", Journal of Geophysical Research: Oceans, 100: 2501-2514.

Castellari S., Pinardi N., Leaman K. (1998), "A model study of air-sea interactions in the Mediterranean Sea", Journal of Marine Systems, 18: 89-114.

Chesbrough, H.W. (2003). Open Innovation: The new imperative for creating and profiting from technology. Boston: Harvard Business School Press. Boston (US).

Dobricic S. and Pinardi N. (2008), "An oceanographic three-dimensional variational data assimilation scheme", Ocean Modelling, 22, 3-4, 89-105

Dobricic S., Adani M., Fratianni C., Bonazzi A. and Fernandez V. (2008), "Daily oceanographic analyses by the Mediterranean basin scale assimilation system", Ocean Sci., 3: 149-157.

Guarnieri A., Oddo P., Pastore M., Pinardi N. (2008), "The Adriatic Basin Forecasting System new model and system development", Coastal to Global Operational Oceanography: Achievements and Challenges. Eds. H. Dahlin, N.C Fleming, and S.E. Petersson. Proceeding of 5th EuroGOOS Conference, Exeter (accepted)

Lecci R., Coppini G., Creti S., Turrisi G., D'Anca A., Palazzo C., Aloisio G., Fiore S., Bonaduce A., Mannarini G., Kumkar Y., Ciliberti S. A., Federico I., Agostini P., Bonarelli R., Martinelli S., Marra P., Scalas M., Tedesco L., Rollo D., Cavallo A., Tumolo A., Monacizzo T., Spagnulo M., Pinardi N., Fazioli L., Olita A., Cucco A., Sorgente R., Tonani M. and Drudi M. "SeaConditions: Present and future sea conditions for safer navigation, OCEANS 2015 Genova, 2015. 10.1109/OCEANS-Genova.2015.7271764.

Oddo P., Pinardi N. and Zavatarelli M. (2005), "A numerical study of the interannual variability of the Adriatic Sea (2000-2002)", Sci. Total Environ., 353: 39-56.

Oddo P., Pinardi N., Zavatarelli M. and Colucelli A. (2006), "The Adriatic Basin forecasting system", , Acta Adriatica, 47(Suppl):169-184.

Oddo P., Adani M., Pinardi N., Fratianni C., Tonani M., Pettenuzzo D. (2009), "A nested Atlantic-Mediterranean Sea General Circulation Model for Operational Forecasting", Ocean Sci. Discuss., 6: 1093-1127

Olita A., Dobricic S., Ribotti A., Fazioli L., Cucco A., Dufau C. and Sorgente R. (2012), "Impact of SLA assimilation in the Sicily Channel Regional Model: model skills and mesoscale features", Ocean Sci., 8: 485–496, doi:10.5194/os-8-485-2012.

Olita A., Sorgente R., Natale S., Gabersek S., Ribotti A., Bonanno A. and Patti B. (2007), "Effects of the 2003 European heat-wave on the Central Mediterranean Sea: surface fluxes and the dynamical response", Ocean Sci., 3: 273–289, doi:10.5194/os-3- 273-2007.

Pinardi N., Allen I., Demirov E., De Mey P., Korres G., Lascaratos A., Traon P.Y., Maillard C., Manzella G. and C. Tziavos (2003), "The Mediterranean ocean forecasting system: first phase of implementation (1998-2001)", Ann. Geophys., 21: 3-20.

Raicich F. (1994), "Note on flow rates of the Adriatic rivers", Technical Report. CNR Istituto Talassografico Sperimentale, Trieste. RF 02/94, 8 pp.

Sorgente R., Drago A.F., Ribotti A. (2003), "Seasonal variability in the Central Mediterranean Sea circulation". Annales Geophysicae, 21: 299-322.

Sorgente R., Olita A., Oddo P., Fazioli L., Ribotti A. (2011), "Numerical simulation and decomposition of kinetic energy in the Central Mediterranean: insight on mesoscale circulation and energy conversion", Ocean Science, 7: 503-519.

Tolman H. (2009), "User Manual and system documentation of WAVEWATCH III version 3.14", NOAA/NWS/NCEP/MMAB Technical Note 276, 194 pp + Appendices

Tonani M., Pinardi N., Dobricic S., Pujol I. and Fratianni C. (2008), "A high-resolution free-surface model of the Mediterranean Sea", Ocean Sci., 4: 1-14

Tonani M., Pinardi N., Fratianni C., Pistoia J., Dobricic S., Pensieri S., de Alfonso M., Nittis K. (2009), "Mediterranean Forecasting System: forecast and analysis assessment through skill scores". Ocean Sci., 5: 649-660

Von Hippel, E. (1986). Lead users: a source of novel product concepts. Management Science 32, 791–805. http://dx.doi.org/10.1287/mnsc.32.7.791.

- sheets of paper in 3D space as the design metaphor;

- cards user interface and the preview/detail pattern;

- bold use of colour, typography, images and animations.

---

## Author Comment (AC2) · 19 Dec 2016

Replies to Referee 2

Thank you very much for your very relevant comments. In the following text we present your comments followed by our answers and the modification that we propose following your comments to the final variant of the manuscript.

Referee's Comment 1 The ms provides the description of a new online, both via web and mobile, user friendly system to access and to visualize the Mediterranean ocean forecasting data provided by : CMEMS, sub-regional ocean systems, ECMWF and other complimentary sea data, in order to assist the end user and to strength the safety

of mariners. SeaConditions is an excellent tool to retrieve and visualize operationally the CMESM met-ocean data in the Mediterranean Sea. However, it was desirable to provide in introduction a short description for 2-3 previous "similar" systems and their disadvantages compared to SeaConditions tool. To my knowledge SeaConditions is well received by end users and not only and the methodology behind it, can be adapted for other down streaming applications, such as oil spill and floating objects predictions.

Authors' answer We thank the Referee for the positive comments and we have followed the referee's suggestion provide in introduction a short description for 3 previous "similar" systems and their disadvantages compared to SeaConditions tool.

We have added the following text to the paragraph: Other software or web portals similar to SeaConditions such as 'Weather4D' and 'Meteomed' do not offer the same integration with marine data. Meteomed does not offer currents and high resolution model data. Both Weather4D and Meteomed do not offer the zoom capacity in the free version. Both of them do not offer bathymetry as one of the product. INGV has developed, since the begin of the Mediterranean Forecasting System (MFS), web interface to visualise the oceanographic products of the Mediterranean Sea (http://medforecast.bo.ingv.it/mfs-copernicus) developed in the different projects (MFSPP, MFSTEP, MYOCEANs and now CMEMS). The INGV website is more scientific oriented respect to SeaConditions, it consists of static images without any zoom capability and does not provide the meteorological forecasts. Other many systems (e.g. Puertos forecasting system portus.puertos.es; CYCOFOS forecasting systems http://www.oceanography.ucy.ac.cy/cycofos/high-resolution.html), most of them part of MONGOOS (http://www.mongoos.eu/in-situ-and-forecasts), provide the access to visualization services of oceanographic forecasting products but all of them for subregional areas only and not for the entire Mediterranean sea, most of them with limited zoom capability and without providing access to atmospheric data. Only few of the forecasting systems in the Mediterranean Sea provide access through mobile applications and only for sub-regional areas of the Mediterranean Sea (IMAR by Puertos

available on Google store for the Western Mediterranean Sea).